# Features of Audio-Vestibular Deficit and 3D-FLAIR Temporal Bone MRI in Patients with Herpes Zoster Oticus

**DOI:** 10.3390/v14112568

**Published:** 2022-11-20

**Authors:** Jiyeon Lee, Jin Woo Choi, Chang-Hee Kim

**Affiliations:** 1Department of Otorhinolaryngology-Head and Neck Surgery, Konkuk University Chungju Hospital, Chungju 27376, Republic of Korea; 2Department of Radiology, Konkuk University Medical Center, Research Institute of Medical Science, Konkuk University School of Medicine, Seoul 05030, Republic of Korea; 3Department of Otorhinolaryngology-Head and Neck Surgery, Konkuk University Medical Center, Research Institute of Medical Science, Konkuk University School of Medicine, Seoul 05030, Republic of Korea

**Keywords:** Ramsay Hunt syndrome, 3D-FLAIR MRI, audio-vestibular deficit, hearing loss, vertigo

## Abstract

Herpes zoster oticus (HZO) is characterized by otalgia and erythematous vesicles in the auricle or external auditory canal. Ramsay Hunt syndrome (RHS) can be diagnosed when facial nerve palsy is accompanied by these symptoms of HZO, and in this case, audio-vestibular symptoms such as hearing loss or dizziness often develop. Recently, 3D-fluid-attenuated inversion recovery sequence (3D-FLAIR) magnetic resonance imaging (MRI) has been introduced in order to evaluate the inner ear structure pathology. The purpose of this study was to investigate the audio-vestibular characteristics in correlation with temporal bone MRI findings in HZO patients. From September 2018 to June 2022, 18 patients with HZO participated in the study. Thirteen patients (77%) showed high-signal intensity in the inner ear structures in 4 h post-contrast 3D-FLAIR images. In a bithermal caloric test, the lateral semicircular canal showed high signal intensity in 4 h post-contrast 3D-FLAIR images in 75% of patients with abnormal canal paresis. While the cochlea showed high signal intensity in 4 h post-contrast 3D-FLAIR images in 75% of patients with hearing loss, the vestibulo-cochlear nerve showed enhancement in post-contrast T1-weighted images in only 33% of patients with hearing loss. The present study demonstrates that audio-vestibular deficits are well-correlated with increased signal intensity of the inner ear endorgans in 4 h post contrast 3D-FLAIR MRI.

## 1. Introduction

Herpes zoster oticus (HZO) is known to be caused by reactivation of varicellea zoster virus (VZV), and it is characterized by otalgia, the vesicular eruptions of the auricle or external auditory canal. Ramsay Hunt syndrome can be diagnosed when ipsilesional facial nerve palsy (FNP) is associated with HZO. The FNP is generated by reactivation of dormant VZV in the geniculate ganglion, and this was confirmed by DNA detection of VZV in the geniculate ganglion [1]. In RHS, symptoms of audio-vestibular deficits, such as hearing loss and vertigo, are commonly accompanied by FNP [2,3,4,5,6,7,8,9]. In addition, paralysis of other cranial nerves can be associated with RHS [10,11,12,13,14,15]. Regarding pathological causes of audio-vestibular deficit, it is controversial whether or not the lesioned sites are inner ear endorgans or vestibulo-cochlear nerves [16,17].

Although the contrast-enhanced 3D-T1-weighted image (CE-3D-T1WI) has been used to identify the facial nerve (CN-VII) in patients with FNP [18,19,20], it is difficult to identify abnormal findings in the inner ear endorgans of CE-3D-T1WI in RHS patients. Recently, 3D-fluid-attenuated inversion recovery sequence (3D-FLAIR) MRI has been widely used as a tool to observe changes in inner ear structure or nerves in RHS patients [9,10,11,12].

The purpose of this study is to investigate how the audio-vestibular test results of RHS patients appear in 3D-FLAIR MRI.

## 2. Materials and Methods

### 2.1. Subjects

The subjects were 18 patients diagnosed with RHS who visited our hospital from September 2018 to June 2022. RHS was diagnosed in patients with vesicles in the external auditory canal or auricle, and CN-VII dysfunction (facial nerve palsy) or vestibulo-cochlear nerve (CN-VIII) dysfunction (hearing difficulty, tinnitus, ear fullness, and dizziness). FNP was confirmed through a physical examination based on the House-Brackmann grading system [21]. All patients underwent pure tone audiometry (PTA). “Hearing loss” was diagnosed when the PTA threshold of speech frequency (0.5 kHz + 1 kHz + 2 kHz/3) or isolated high frequency (4 kHz + 8 kHz/2) was 10 dB or more than that of a healthy ear [13]. Under an infrared video-based system (CHARTR VNG, ICS Medical, Schaumburg, IL, USA), a bithermal caloric test was carried out with alternating irrigation of 30 °C and 44 °C water for 30 s. Canal paralysis (CP) was determined using the Jongkees formula [22]. Vestibular-evoked myogenic potential (VEMP) testing was performed using an auditory evoked potential system (Navigator Pro^®^, Bio-logic Systems Corp, Mundelein, IL, USA). Abnormal VEMPs were defined as absent reflexes without recognizable tracking. The video head impulse test (vHIT) was performed using an ICS Impulse device (Natus Medical Inc., Taastrup, Denmark). Laboratory testing of VZV was performed using antibody titer or PCR assays.

### 2.2. MRI

All patients underwent a temporal bone MRI, using a 3.0 T MRI (Signa HDx: GE Healthcare, Milwaukee, WI, USA) with a phased-array head coil. T1WI and T2 FLAIR were taken as axial thin sections using 3D imaging techniques, including the bilateral temporal bone, inner ear structure, cerebellum, brainstem, and the base of the temporal lobe. Based on this, axial and coronal reconstruction images were obtained. The contrast medium was intravenously injected with a standard dose of gadobutrol (Gadovist; Schering, Berlin, Germany; 0.1 mmol/kg body weight) at a rate of 1 mL per second, using a power injector. After contrast injection, using 3D imaging techniques, T1WI and T2 FLAIR were repeatedly photographed 10 min, 15 min, and 4 h after the axial thin sections, in order to obtain axial images and coronal reconstruction. The enhancement of the structures was confirmed by a radiologist.

## 3. Results

A total of 18 patients (9 men and 9 women; mean age 52.8 year; range, 19–89) were enrolled in this study. The side of the lesion was the right in 10 cases and the left in 8 cases. One of the subjects did not perform vHIT or cervical VEMP (cVEMP) tests (patient No. 1). Table 1 shows the audio-vestibular results and the features of the cochlea, lateral semicircular canal (LSCC), posterior semicircular canal (PSCC), vestibule, CN-VII, and CN-VIII in post-contrast 3D-FLAIR MRI or CE-3D-T1WI.

In post-contrast 3D-FLAIR, 13 patients showed high-signal intensity of the inner ear structure on the lesion side compared to the unaffected side (Figure 1a). The proportion of patients for whom CE-3D-T1WI was enhanced was not significantly different between patients with and without auditory-vestibular deficits (*p* = 0.278, Fisher’s exact test).

Eleven patients showed FNP. Among them, ten patients showed higher signal intensity of the facial nerve than that of the unaffected side in CE-3D-T1WI (Figure 1b). Of the seven patients without FNP, three showed high-signal intensity of the facial nerve at CE-3D-T1WI. The proportion of the patients with enhancement in CE-3D-T1WI was significantly higher in patients with FNP (*p* = 0.047, Fisher’s exact test).

Twelve patients showed abnormal caloric response (canal paresis >25%) on the lesion side. Among them, nine patients (75%) showed high-signal intensity of LSCC in post-contrast 3D-FLAIR. Six patients showed decreased vHIT gain of the LSCC or PSCC. All six patients with reduced vHIT gain of LSCC showed high-signal intensity of LSCC in post-contrast 3D-FLAIR. All three patients with decreased vHIT gain of PSCC showed high-signal intensity of PSCC in post-contrast 3D-FLAIR. However, even in patients with normal vHIT gain, PSCC, in four patients, and LSCC, in five patients, showed high signal intensity in post-contrast 3D-FLAIR. Of the twelve patients with hearing loss in PTA, nine showed high-signal intensity of cochlea in post-contrast 3D-FLAIR. In four patients, high-signal intensity of CN-VIII was observed in CE-3D-T1WI (Figure 1c). In ten patients with abnormal VEMP, eight patients showed high-signal intensity in the vestibule. Among the seven patients with VEMP response, five patients showed high-signal intensity of the vestibule.

## 4. Discussion

Audio-vestibular deficits are commonly associated with HZO [3]. It has traditionally been known that because VZV tends to invade the facial nerve, vestibular nerve, and cochlear nerve, in that order, the CN-VIII is rarely involved without FNP in RHS [23]. However, the present study showed that 71% (5 of 7) of HZO patients without FNP complained of dizziness and/or hearing loss (Table 1). Among patients with FNP (*n* = 11), ten patients (91%) showed enhancement of the CN-VII in CE-3D-T1WI, and one patient (9%) showed no enhancement. Among patients without FNP (*n* = 7), three patients (43%) showed enhancement of the CN-VII in CE-3D-T1WI, and four patients (57%) showed no enhancement. The proportion of patients with enhancement in CE-3D-T1WI was significantly higher in patients with FNP (*p* = 0.047, Fisher’s exact test), which is consistent with previous observations [20,24]. Minakata et al. observed significant correlations between facial nerve enhancement in CE-3D-T1WI and facial nerve swelling in the labyrinthine segment, geniculate ganglion, and pyramidal segment in RHS patients who underwent decompression surgery [24]. They also reported that the patients with lower contrast enhancement in the tympanic and mastoid segments demonstrated better prognoses for facial nerve recovery after surgery [24]. Kuya et al. observed facial nerve enhancement in both post-contrast 3D-FLAIR and CE-3D-T1WI in all five patients with RHS [25].

Among patients with symptoms of audio-vestibular deficits (*n* = 14), five patients (36%) exhibited enhancement of the CN-VIII in CE-3D-T1WI, and nine patients (64%) exhibited no enhancement. Among patients without symptoms of audio-vestibular deficits (*n* = 4), none (0%) exhibited enhancement of the CN-VIII in CE-3D-T1WI. The proportion of patients with enhancement in CE-3D-T1WI was not significantly different between the patients with an audio-vestibular deficit and those without one (*p* = 0.278, Fisher’s exact test). While the proportion of the patients with symptoms of audio-vestibular deficits among those who showed enhancement of the CN-VIII in CE-3D-T1WI was high (100% sensitivity, five of five), the proportion of patients without symptoms of audio-vestibular deficits among those who did not show enhancement of the CN-VIII in CE-3D-T1WI was relatively lower (31% specificity, four of thirteen). On the other hand, a 4 h post-contrast 3D-FLAIR MRI demonstrated different results. Among patients with symptoms of audio-vestibular deficits (*n* = 14), twelve patients (86%) showed enhancement of the inner ear endorgans in 4 h post-contrast 3D-FLAIR images, and two patients (14%) showed no enhancement. Among patients without symptoms of audio-vestibular deficits (*n* = 4), three patients (75%) showed no enhancement of the inner ear endorgans in 4 h post-contrast 3D-FLAIR images (Table 1). The proportion of the patients with enhancement of the inner ear endorgans in 4 h post-contrast 3D-FLAIR MRI images was significantly different between the patients with an audio-vestibular deficit and those without one (*p* = 0.044, Fisher’s exact test).

An abnormal CP, in a bithermal caloric test, was observed in 12 patients (of 18, 67%). Of twelve patients with an abnormal CP, nine patients (75%) showed enhancement of the LSCC in a 4 h post-contrast 3D-FLAIR MRI. Of six patients without an abnormal CP, four patients (67%) showed no enhancement of the LSCC in a 4 h post-contrast 3D-FLAIR MRI. Although the patients with abnormal CP tended more commonly to show enhancement of the LSCC, the proportion of the patients with enhancement of the LSCC according to the 4 h 3D-FLAIR MRI was not significantly different between the patients with and without an abnormal CP (*p* = 0.141, Fisher’s exact test). Of twelve patients with an abnormal CP, four patients (33%) showed enhancement of the CN-VIII in CE-3D-T1WI, while of six patients without an abnormal CP, one patient (17%) showed enhancement of the CN-VIII in CE-3D-T1WI. The proportion of the patients with enhancement of the CN-VIII in CE-3D-T1WI was not significantly different between the patients with and without an abnormal CP (*p* = 0.615, Fisher’s exact test).

Most of the patients with decreased vHIT gain of LSCC or PSCC showed an enhancement of the corresponding lesion in 4 h post-contrast 3D-FLAIR images. However, even in patients with normal vHIT gain, signal changes were observed in LSCC or PSCC in some cases. In 4 h post-contrast 3D-FLAIR images, the signal change in the LSCC seems to be better represented by the abnormal CP in the bithermal caloric test than by the vHIT gain decrease.

Regarding hearing loss, 75% of patients with hearing loss showed enhancement of the cochlea in post-contrast 3D-FLAIR images, and 33% showed enhancement of the CN-VIII in CE-3D-T1WI. Berrettini et al. conducted a study on 3D-FLAIR images in patients with idiopathic sudden sensorineural hearing loss [26]. They reported that thirteen of twenty-three patients showed high signal intensity in the inner ear structures in pre-contrast 3D-FLAIR images, of which seven patients (of thirteen) also showed post-contrast enhancement. Among thirteen patients, eleven patients showed high signal intensity in the cochlea, and three patients showed high signal intensity in the cochlear nerve [26].

Sugiura et al. reported a case with RHS accompanying vertigo, showing high signal intensity in the cochlea and vestibule in pre-contrast 3D-FLAIR images without enhancement in 3D-T1WI [27]. Nakata et al. studied the 3D-FLAIR findings of the inner ear structures in patients with Bell’s palsy and RHS [28]. Two out of three patients with RHS showed high signal intensity in the inner ear structures (one in the cochlea and one in the vestibule, respectively) in pre-contrast 3D-FLAIR images. They reported no enhancement of the inner ear structures in post-contrast 3D-FLAIR images in these patients. Kuya et al. investigated temporal bone MRI findings of five patients with RHS [25]. Two patients showed high signal intensity in pre- and post-contrast 3D-FLAIR images, and all five patients showed high signal intensity of CN-VII in pre- and post-contrast 3D-FLAIR images and CE-3D-T1WI. The CN-VIII exhibited high signal intensity for the pre-contrast 3D-FLAIR images of two patients, and showed enhancement of the post-contrast 3D-FLAIR images in four patients.

The present study demonstrated that the symptoms of audio-vestibular deficits in HZO patients more relevantly correspond to the enhancement of the inner ear structures in 4 h post-contrast 3D-FLAIR images than to the enhancement of CN-VIII in CE-3D-T1WI. Furthermore, in RHS patients, the enhancement of the LSCC in 4 h post-contrast 3D-FLAIR images is thought to be related to the abnormal caloric CP rather than a vHIT gain decrease, and it does not seem to be significantly related to enhancement of CN-VIII in CE-3D-T1WI.

## 5. Conclusions

The present study demonstrated that the symptoms of audio-vestibular deficits in HZO patients more relevantly correspond to the enhancement of the inner ear structures in 4 h post-contrast 3D-FLAIR images than to the enhancement of CN-VIII in CE-3D-T1WI. Furthermore, an abnormal caloric CP and a vHIT gain decrease are more relevantly represented by enhancement of the corresponding inner ear structures in 4 h post-contrast 3D-FLAIR images than enhancement of the CN-VIII in CE-3D-T1WI.

## Figures and Tables

**Figure 1 viruses-14-02568-f001:**
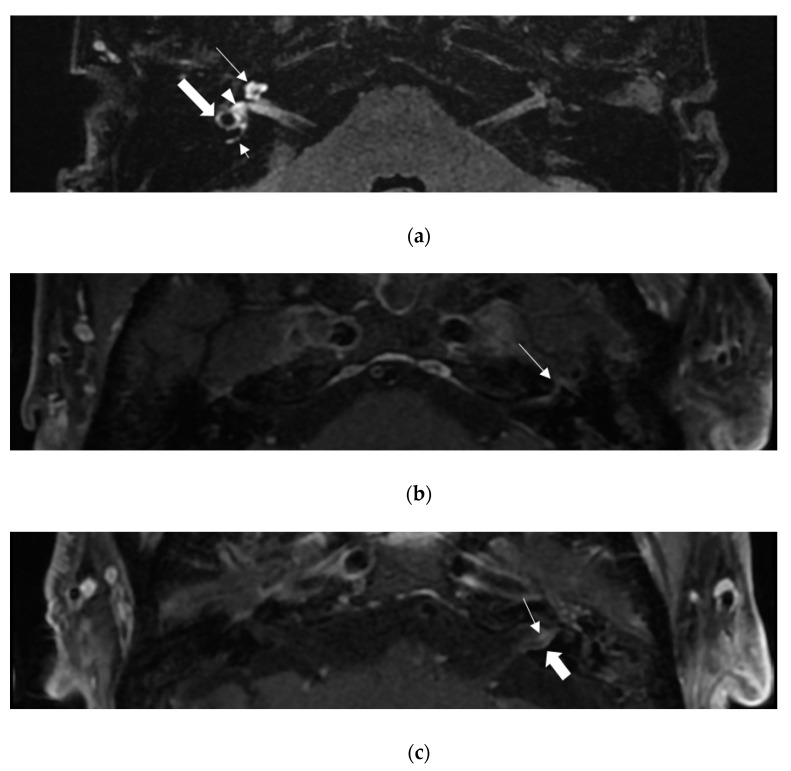
Internal auditory canal (IAC) MRI images of patients with Ramsay Hunt syndrome. Post-contrast 3D-FLAIR MRI of patient 12 (**a**), and post-contrast 3D T1-weighted images of patient 13 (**c**) and patient 9 (**b**). (**a**) High-signal intensities of cochlea (long arrow), posterior semicircular canal (small arrow), lateral semicircular canal (large arrow), and vestibule (arrowhead) were observed in the right ear. (**b**) The labyrinthine segment of the left facial nerve is enhanced (arrow). (**c**) In the left ear, contrast enhancement is observed in the vestibulo-cochlear nerve(small arrow) and the dura of the IAC (large arrow), respectively.

**Table 1 viruses-14-02568-t001:** Clinical features and MRI findings of patients with Ramsay Hunt syndrome (*n* = 18).

Patient No.	Sex/Age	Lesion Side	Vestibulo-Cochlear Symptom	FNP	CP (%)	Decreased vHIT Gain	cVEMP	Hearing Loss	PTA (Speech Frequency/High Frequency)	CE-3D-T1WI	Post-Contrast 3D-FLAIR
1 *	M/39	R	-	−	Rt 30 ^†^			−	11/27	-	-
2	M/42	L	Dizz EF Tinn	+	Lt 43	-	A ^‡^	+	16/30	VII	C V
3	F/22	R	Dizz EF HL	+	Lt 7	-	A	+	23/20	-	-
4	M/47	L	Dizz	+	Lt 68	-	A	+	30/97	VII	C V LSCC PSCC
5	F/66	R	Dizz HL	+	Rt 100	LSCC	A	+ ^§^	40/60	VII	C V LSCC PSCC
6	F/44	R	Dizz	−	Rt 31	-	A	−	20/22	-	C V LSCC PSCC
7	F/46	R	-	+	Lt 1	-	N ^‖^	−	13/37	VII	-
8	F/59	L	-	+	Rt 13	-	N	−	15/17	VII	-
9	M/76	L	Dizz	+	Lt 82	PSCC LSCC	A	+	61/77	VIII	C V LSCC PSCC
10	F/35	R	Dizz EF HL	+	Rt 48	-	A	+	18/30	VII VIII	C V LSCC PSCC
11	M/71	L	EF HL	−	Lt 21	-	A	+	95/85	-	V
12	F/60	R	Dizz HL	−	Rt 100	PSCC LSCC	N	+	71/92	VII	C V LSCC PSCC
13	M/89	L	-	−	Lt 20	LSCC	A	−	71/so	VII	C V LSCC PSCC
14	M/64	R	Dizz EF HL Tinn	−	Rt 63	-	A	+	33/42	-	-
15	F/60	L	Dizz	+	Lt 64	LSCC	N	+	21/37	VII VIII	C V LSCC
16	M/19	L	Dizz	+	Lt 24	-	N	−	15/12	VII VIII	C V LSCC PSCC
17	M/57	R	Dizz	−	Rt 92	PSCC LSCC	N	+	80/so	VII VIII	C V LSCC PSCC
18	F/55	R	Dizz	+	Rt 34	-	N	+	21/42	VII	C V LSCC

C = cochlea; CP = canal paresis; Dizz = dizziness; EF = ear fullness sensation; F = female; HL = hearing loss; LSCC = lateral semicircular canal; M = male; PSCC = posterior semicircular canal; so = scale out; Tinn = tinnitus; V = vestibule; vHIT = video head impulse test; VII = facial nerve; VIII = vestibulocochlear nerve. * This patient has not been tested for vHIT and cVMEP. ^†^ Rt or Lt indicates that canal paresis was observed in the right (Rt) or left (Lt) side. ^‡^ Abnormal in the lesioned side. ^§^ This patient had profound hearing loss on the contralesional side. ^‖^ Normal in the lesioned side.

## Data Availability

Not applicable.

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
