# Peer review of "Features of Audio-Vestibular Deficit and 3D-FLAIR Temporal Bone MRI in Patients with Herpes Zoster Oticus"

_viruses, 2022, doi:10.3390/v14112568_

Round 1

Reviewer 1 Report

Line 20 – delete “of 18”

Line 56 – add reference for “House-Brackmann grading”

Line 63 – add reference for “Jongkees formula”

Lines 58-60 – “as follows: PTA threshold of speech frequency (…) or isolated frequency (…) at the affected ear is 10 dB or more…”

Lines 61-62 – delete 2nd repetition “bithermal caloric test was”

Line 82 – “the side of the lesion was right in 10 cases and left in 8”

All statistical analysis (and possible significances) are reported in “discussion”, while I would recommend to anticipate them also in “Results” chapter [e.g. lines 89-92]; [e.g. shift lines 153-154 in “results”]; [same for lines 165-167, anticipate in “results] and so on 

Lines 111-113 – what about vestibule signal in patients with normal VEMPS? Additionally, no further reference to VEMPS (normal/abnormal) and MRI vestibule signal in the paper

Lines 168-172 – could be clearer by replacing “that showed enhancement” with “among the ones that showed enhancement”; same for line 170

Lines 184-187 – repetition of 177-180; delete the second repetition

Line 192 – “And” ? “While”

Line 201 – “Regarding” starts a new paragraph

Line 206 – Delete “And”

Author Response

Thank you for your thoughtful comments, and please review the following revisions.

Line 20 – delete “of 18”

 It has been modified as follows: Thirteen patients (77%) showed high-signal intensity in the inner ear structures in 4hr postcontrast 3D-FLAIR images.

Line 56 – add reference for “House-Brackmann grading”

Line 63 – add reference for “Jongkees formula”

 We added a reference to each.

Lines 58-60 – “as follows: PTA threshold of speech frequency (…) or isolated frequency (…) at the affected ear is 10 dB or more…”

 It has been modified as follows: When the PTA threshold of speech frequency (0.5 kHz + 1 kHz + 2 kHz/3) or isolated high frequency (4 kHz + 8 kHz/2) is 10 dB or more than that of a healthy ear [13].

Lines 61-62 – delete 2nd repetition “bithermal caloric test was”

 We removed the repeating part and modified it as follows: It was with alternating irrigation of 30 °C and 44 °C water for 30s. Canal paralysis (CP) was determined using the Jongkees formula [22].

Line 82 – “the side of the lesion was right in 10 cases and left in 8”

 It has been modified as follows: The side of the lesion was right in 10 cases and left in 8 cases.

All statistical analysis (and possible significances) are reported in “discussion”, while I would recommend to anticipate them also in “Results” chapter [e.g. lines 89-92]; [e.g. shift lines 153-154 in “results”]; [same for lines 165-167, anticipate in “results] and so on 

 The following sentences have been added to the “Results” chapter:

The proportion of patients for whom CE-3D-T1WI was enhanced was not significantly different between patients with and without auditory-vestibular deficits (p = 0.278, Fisher's exact test).

 The proportion of the patients with enhancement in CE-3D-T1WI was significantly higher in patients with FNP (p = 0.047, Fisher’s exact test).

Lines 111-113 – what about vestibule signal in patients with normal VEMPS? Additionally, no further reference to VEMPS (normal/abnormal) and MRI vestibule signal in the paper

It has been modified as follows: Among the 7 patients with VEMP response, 5 patients showed high-signal intensity of the vestibule.

Lines 168-172 – could be clearer by replacing “that showed enhancement” with “among the ones that showed enhancement”; same for line 170

 It has been modified as follows: While the proportion of the patients with symptoms of audio-vestibular deficit among the ones that showed enhancement of the CN-VIII in CE-3D-T1WI was high (100% sensitivity, 5 of 5), the proportion of the patients without symptoms of audio-vestibular deficit among the ones that did not show enhancement of the CN-VIII in CE-3D-T1WI was relatively lower (31% specificity, 4 of 13).

Lines 184-187 – repetition of 177-180; delete the second repetition

 We removed the repeating part.

Line 192 – “And” ? “While”

 It has been modified as follows: While of 6 patients without an abnormal CP, 1 patient (17%) showed enhancement of the CN-VIII in CE-3D-T1WI.

Line 201 – “Regarding” starts a new paragraph

Line 206 – Delete “And”

 We've revised the paragraph based on your comments and removed "And".

Reviewer 2 Report

THis manuscript can be improved if the correlation of the Vestibulo-cochlear symptoms, the averaged PTA threshold and the severity and/or number of vestibular organs is further analyzed.

Author Response

Thanks for your thoughtful comments. The correlation between symptoms and vestibular organs in 3D-FLAIR MRI was mentioned in other previous papers, so we did not analyze it. We modified the PTA threshold in the table by dividing it into speech frequency and high frequency.

Reviewer 3 Report

This study investigated the relationship between features of audio-vestibular deficit and findings in 3D-FLAIR MRI in patients with herpes zoster oticus. The topic itself deserves to be addressed; however, this study has serious flaws.

-        Were the subjects patients with Ramsay Hunt syndrome or herpes zoster oticus? There seems to be confusion throughout the manuscript.

-        In figure 1B, enhancement of the facial nerve is not apparent. The authors did not mention how they decided whether the target area is enhanced or not, and the reliability of their decision might be suspected.

-        The degree of facial palsy and hearing loss should be presented quantitatively.

Author Response

Herpes zoster oticus was known in 1904 by German otologist Körner as a disease that specifically caused vesicles in the auricle, facial nerve palsy, and dysfunction of the inner ear. After that, Hunt was the first to describe and analyze the diversity of clinical manifestations of herpes zoste oticus, and for his achievements in which inflammation invades the facial nerve and causes neurological dysfunction, it is now called Ramsay Hunt syndrome after him. In general, when peripheral facial palsy, otalgia, and ear vesicles are accompanied, it is called Ramsay Hunt syndrome, but it is sometimes used interchangeably.

In this study, the enhancement of the structure was determined through the supervision of a radiologist. Facial nerve enhancement in 1B has also been confirmed by radiologists. We added the following sentence to the "Materials and Methods" chapter: The enhancement of the structures was confirmed by a radiologist.

The degree of facial palsy was not recorded in all enrolled patients at the time of the first visit, so the degree of paralysis could not be explained. Hearing loss was expressed quantitatively and the table was modified. Please see the attachment.

Reviewer 4 Report

This is interesting paper about the MRI findings in patients with HZO.

The main problem with the manuscript is that there is not any explanation about the findings: 1) what is the contribution of the different test to understand the HZO mechanisms?. 2) what does the "enhacement" in the MRI mean, why did they find different results after 4 hours?, 3) what is the relationship between the tests results and MRI finding?

Apart from this the discussion is very difficult to follow, tables or other more didactic method is mandatory for the readers to present the results.

Author Response

Thanks for your detailed comments. According to a previous study, 4-hour delayed-enhanced images were dominant in neural inflammatory-dominant conditions in patients with unilateral ear symptoms [1]. Our results also observed the strongest enhanment of inner ear structures in the images after 4 hours. Test results correlate better in 3D-FLAIR than in CE-3D-T1WI for inner ear structures.

[1] T.Y. Kim, D.W. Park, Y.J. Lee, J.Y. Lee, S.H. Lee, J.H. Chung and S. Lee. Comparison of Inner Ear Contrast Enhancement among Patients with Unilateral Inner Ear Symptoms in MR Images Obtained 10 Minutes and 4 Hours after Gadolinium Injection. American Journal of Neuroradiology December 2015, 36 (12) 2367-2372; DOI: https://doi.org/10.3174/ajnr.A4439

Round 2

Reviewer 4 Report

In this correction, the authors did not change my major concerns. In the paper, they use imaging where they found changes in "enhancements" without any explanation about the meaning of this finding. To evaluate the patient´s conditions they use current tests in Neurotology: vHIT, VEMPs, and Audiograms, again there are no explanations about the findings and their consequences, and of course, there is not a clear relationship between the clinical, imaging, and ancillary methods.

The discussion is impossible to follow for the reader, a table, chart, etc is mandatory to show the findings.

In its current form this paper it is not useful for Viruses readers.